# The Evolution of Pharmacological Activities *Bouea macrophylla* Griffith In Vivo and In Vitro Study: A Review

**DOI:** 10.3390/ph15020238

**Published:** 2022-02-16

**Authors:** Intan Tsamrotul Fu’adah, Sri Adi Sumiwi, Gofarana Wilar

**Affiliations:** Department of Pharmacology and Clinical Pharmacy, Faculty of Pharmacy, Universitas Padjadjaran, Jl. Raya Bandung-Sumedang, Jatinangor, Sumedang 45363, West Java, Indonesia; intan19018@mail.unpad.ac.id (I.T.F.); sri.adi@unpad.ac.id (S.A.S.)

**Keywords:** *Bouea macrophylla* Griffith, antioxidant, anticancer, antihyperglycemic, antimicrobial, antiphotoaging

## Abstract

*Bouea macrophylla* Griffith (*B. macrophylla)* is one of the many herbal plants found in Asia, and its fruit is plum mango. This plant is rich in secondary metabolites, including flavonoids, tannins, polyphenolic compounds, and many others. Due to its bioactive components, plum mango has powerful antioxidants that have therapeutic benefits for many common ailments, including cardiovascular disease, diabetes, and cancer. This review describes the evolution of plum mango’s phytochemical properties and pharmacological activities including in vitro and in vivo studies. The pharmacological activities of *B. macrophylla* Griffith reviewed in this article are antioxidant, anticancer, antihyperglycemic, antimicrobial, and antiphotoaging. Each of these pharmacological activities described and studied the possible cellular and molecular mechanisms of action. Interestingly, plum mango seeds show good pharmacological activity where the seed is the part of the plant that is a waste product. This can be an advantage because of its economic value as a herbal medicine. Overall, the findings described in this review aim to allow this plant to be explored and utilized more widely, especially as a new drug discovery.

## 1. Introduction

*B. macrophylla* Griffith is a species of the Anacardiaceae family, having a fruit similar to mango (*Mangifera indica* belong to the same family (Anacardiaceae)), known as plum mango. This species has various names in various regions, such as gandaria (Indonesia and the Philippines), maprang or mayong (Thailand), and kundang or ramania (Malaysia) [1,2,3,4]. This plant grows to 27 m, the stem is grayish brown, and the leaves are dark green with a diameter of 13–15 cm (Figure 1A), the fruit is yellowish-green (Figure 1B), and the seed is purple (Figure 1C) [4]. Plum mango has seeds with purple cotyledons, one of its distinctive characteristics. The tree is widely cultivated in Asia, including Indonesia, Thailand, Malaysia, and India. This species grows in the tropics and thrives below an altitude of 300 m, and are called tropical fruits. Plum mango has been used as herbal medicine to cure headaches, diabetes mellitus, and as an antibiotic mouthwash to treat canker sores [5].

This plant is reported to have many benefits from every part of the plant, including the fruit, leaves, stems, and even the seeds of the fruit. In addition, it is also widely consumed by the community; the leaves are used as fresh vegetables and the fruit is eaten *rujak* or juiced for daily consumption in Indonesia. The community also uses the bark to make agricultural and household tools [5]. The use of this plant is due to its nutritional content, which includes protein, fiber, carbohydrates, vitamins, and the content of its phytochemical compounds; the phytochemical compounds include flavonoids, phenols, saponins, tannins, terpenoids, and steroids [5,6]. However, it seems polyphenolic compounds, including phenol, flavonoid, and tannin, are the most valuable compound related to their pharmacological activity [7,8,9].

Some species of the Anacardiaceae family that have the main bioactive compounds polyphenolic compounds include *Mangifera indica* (mango), *Rhus coriaria* (sumac), and *Anacardium occidentale* (cashew). They are reported to have various pharmacological activities such as antioxidant, antimicrobial, anticancer, and antihyperglycemic [10,11,12,13,14]. Related to these data, plum mango has been reported to have various pharmacological activities, including antioxidant [2,3,15,16,17,18,19,20,21,22], anticancer [3,23,24,25,26,27], antimicrobial [3,28,29], antihyperglycemic [15,16,20,30], antiphotoaging [30], and can increase vegetable intake and increase the concentration of beta carotene in the blood [31].

By virtue of the use of *B. macrophylla* Griffith as a herbal medicine and the lack of widespread use of this plant due to the limited information about its therapeutic benefits and compound content, we present this review of the pharmacological activities of *B. macrophylla* Griffith. In this review, each of the pharmacological activities of *B. macrophylla* Griffith will be reviewed, including the source of plant parts, their activities and mechanisms, and their relationship to the secondary metabolic content of this plant.

## 2. Taxonomical Classification

*B. macrophylla* Griffith known as plum mango has another name, *B. gandaria* Blume ex Miq. and *Tropidopetalum javanicum* Turcz. This plant has a common name, gandaria, but has its own name in each region, where the English names are marian plum and plum mango, while in Indonesia it includes: wetes (North Sulawesi), buwa melawe (Sulawesi), barania (Kalimantan), luber (North Sulawesi), Flores), remie (Gajo), gandaria (Java), ramen (Sumatra), pao pandaria (Madura), kalawasa (Makassar), rapo-rapo kebo (Sulawesi), gandoriah (Minangkabau), Gunarjah, jatake, jantake, and kendara (West Java). Then in other Asian countries such as Malaysia, it is known as asam suku, kondongan, kedungan hutan, kundang, kundang medang asam, pako kundang, rembunia, remenya, rumenia, rumia, serapoh, serapok, setar; while in the Philippines it is known as gandaria, and in Thailand known as mapraang, mayong, and somprang [4].

The taxonomical classification of plum mango [32]:
Kingdom: PlantaeDivision: MagnoliophytaClass: MagnolipsidaOrder: SapindalesFamily: AnacardiaceaeGenus: BoueaSpecies: *Bouea macrophylla* (Griffith)

## 3. Nutrient Content of *Bouea macrophylla* Griffith

*B. macrophylla* Griffith or its fruit plum mango is a plant widely cultivated in Indonesia, Thailand, Malaysia, and several other countries [5]. This plant is quite large and tall, similar to a mango tree, a species of the same family. The parts of this plant have been widely used by the community, such as the wood, which is used to make various handicrafts [33]. In addition, the unripe fruit, which tends to have a sour taste, is consumed as rujak, the ripe fruit is made as juice and syrup, the leaves are used as salad, and the seeds are widely used as traditional medicine [5].

Studies on the nutritional content of plum mango have been carried out, including proteins, fats, carbohydrates, minerals, vitamins, and amino acids [6]. The study included both ripe and unripe fruit of plum mango. Based on the study, unripe fruit showed superior content in almost all nutritional parameters than ripe fruit. Among them is a high amino acid content correlated with high protein content. The fiber content of unripe fruit is virtually double that of ripe fruit. In addition, the sugar content of ripe fruit was reported to be higher than that of unripe fruit, especially the content of glucose, fructose, and sucrose [6]. The methanol extract of ripe plum mango has the highest sucrose content, while the most elevated glucose and fructose content is found in the distilled water extract. This is related to the high solubility of simple sugars and oligosaccharides in methanol and water [6].

## 4. Phytochemical Constituents of *Bouea macrophylla* Griffith

Parts of the plant *B. macrophylla* Griffith have been reported to have high antioxidant activity. This is due to the high content of antioxidant compounds such as polyphenols, tannins, flavonoids, and ascorbic acid in a phytochemical screening study [19]. Studies on phytochemical screening of *B. macrophylla* Griffith have been carried out on the fruit, seed, stem, and leaves with various solvents. According to the results of these studies, it can be concluded that water and methanol as solvents are best used to extract the active compounds from this plant.

Ripe fruits are rich in oxygenated compounds, terpenic hydrocarbons (29.28%), ketones (27.27%) and esters (20.73%), acetophenone (12.31%), and acetylvaleryl (10.99%), as the main links identified. While in immature fruits, terpenic hydrocarbons predominate (32.89%) and their subclasses such as α-cadinol (4.94%), α-murools (1.14%), deltacadines (4.80%), and Cameron (3.65%) and acids (29.72%). Aromatic compounds such as eugenol (0.12%), myristic acid (0.34%), α- and β-terpineol (4.41 and 0.09%), thymol (0.55%), octane (0.10%) and ketones such as acetophenone (2.59%) and 5.6 decandione (13.99%) have also been identified in green fruits [34].

The bioactive compounds of plum mango seed identified are gallotanin, a class of phenolic compounds. These compounds are gallic acid (Figure 2A), ethyl gallate (Figure 2B), and pentagalloyl glucose (Figure 2C), which have been reported as the main compounds that play a role in various pharmacological activities of plum mango, but mainly anticancer [3,25,26]. Some other plant parts of plum mango still have limited information about specific bioactive compounds. The research is still ongoing regarding phytochemical screening covering common groups of compounds, but various pharmacological activities have been widely reported.

## 5. Pharmacological Activities of *Bouea macrophylla* Griffith

Based on the data from the research articles that we have collected regarding the various pharmacological activities of *B. macrophylla* Griffith, we have found five activities, namely antioxidant, anticancer, antimicrobial, antihyperglycemic, and antiphotoaging, as well as health benefits by increasing vegetable intake and blood β-carotene concentration. Figure 3 belows summarized overall the pharmacological activities of *B. macrophylla* Griffith.

### 5.1. Antioxidant

Antioxidants are compounds that can ward off free radicals and prevent the harmful effects of free radicals on the body [35]. Free radicals are unstable and reactive molecules because they have one or more unpaired electrons [36]. It can stabilize by gaining or losing an electron, leading to a chain reaction, which ultimately damages vital biological substances like proteins, nucleic acids, and lipids [37]. Free radicals are harmful due to oxidative stress, which is associated with many diseases, including carcinogenesis, cardiovascular disease, diabetes mellitus, and atherosclerosis [38].

Plants are widely known and have been scientifically proven to be the primary sources of antioxidants substances. Natural product metabolites such as flavonoids and phenolics have been shown to possess potent antioxidant activity [39]. *B. macrophylla* Griffith reported high phenolic compound levels that have possible antioxidant activity, as described in Table 1. The antioxidant activity of *B. macrophylla* Griffith has been explored in all parts of the plant, demonstrating that this species’ seed [3,40], fruit [2,15,19], and leaves [17,18,20,21,22] have antioxidant activity. Although all the experiments are in vitro, these findings give essential information about possible mechanisms of *B. macrophylla* Griffith. The most common method used is the DPPH assay, an in vitro assay to measure the radical scavenging effects, of which DPPH is a free radical molecule. The presence of hydrogen atoms from antioxidant compounds that bind to free electrons in radical compounds cause a change from free radicals (diphenylpicrylhydrazyl) to non-radical compounds (diphenylpicrylhydrazine) [41], characterized by a color change from purple to yellow (as antioxidants reduce free radical compounds) [42]. IC50 is used to determine the antioxidant activity using the DPPH method; the concentration of the sample required to capture DPPH radicals is as much as 50% [43]. The greatest IC50 of plum mango using the DPPH method is its seed chloroform extract with 4.34 µg/mL; this IC50 value correlates to its phenolic content, which is the highest compared to other part plant and extraction methods, as represented in Table 1.

Ascorbic acid, also known as vitamin C, is a potent antioxidant and is commonly used as a comparison. The IC50 value of ascorbic acid in its inhibition of DPPH is around 5 µg/mL. The highest IC50 value of mango plum (4 µg/mL) can be compared with ascorbic acid, so it can be concluded that the plum mango seeds extract is a powerful antioxidant. These results show that plum mango can be a candidate for a potent antioxidant. With a very high total phenolic content of 686.05 mg GAE/g, it can be assumed that the possible secondary metabolite compounds that play a role in the antioxidant activity of plum mango are phenolic compounds.

Besides DPPH assay, other antioxidant assays include FRAP, ABTS, and FIC. Determination of antioxidant activity with the ABTS method is to remove the color of the ABTS cation to measure the antioxidant capacity that reacts directly with the ABTS cation radical [44]. ABTS is a radical with a nitrogen center with a characteristic blue-green color. When reduced by antioxidants, it will change to a non-radical form from color to colorless [45]. Meanwhile, the FRAP method is used when carried out on antioxidant compounds that can reduce ferri-tripy-ridyl-triazine (Fe(III)TPTZ) to Ferro-tripyridyl-triazine (Fe(II)TPTZ) complexes [46].

Moreover, the FIC assay is performed to determine the ability of the extracts to chelate ferrous ions. The antioxidant properties of the natural plant extracts can be from their ability to chelate transition metal ions, especially Fe^2+^ and Cu^2+^ [47]. For example, the complex formation between ferrozine and Fe^2+^ can be disturbed by other complexing agents, which causes a decrease in the red color intensity of complexes [48].

Two research articles reported antioxidant activity of *B. macrophylla* Griffith seeds, one of them observed its seeds from ripe and unripe fruit. The IC50 values varied from 4.73 to 46.34 µg/mL based on the DPPH assay [3,40]. Antioxidant activity of *B. macrophylla* Griffith seed with a decoction technique extraction showed a greater IC50 of 4.73 µg/mL than Vitamin C, which has an IC50 of 5.89 µg/mL and is known as a potent antioxidant. This is correlated with total phenolic content, which the seed extract obtained from a decoction technique exhibiting a higher total phenolic 689.17 mg GAE/g [40]. The highest total phenolic content of seed extract with ethanol maceration technique was 550 mg GAE/g [3]. It can be concluded that the best extraction technique for *B. macrophylla* Griffith seed to obtain great antioxidant activity is a decoction.

The other plant part of *B. macrophylla* Griffith that possesses antioxidant activity are the leaves. Some research articles reported that *B. macrophylla* Griffith leaves extract has antioxidant activity; the IC50 values varied with different extraction techniques and antioxidant assays. The research of *B. macrophylla* Griffith leaves widely observed in different stages of maturity, compared different extraction techniques and several extraction solvents. The greatest IC50 of *B. macrophylla* Griffith leaves extract was 2.6 µg/mL with the DPPH assay. The extraction method used was a vacuum evaporator in which the leaves were made into juice, powdered by a vacuum evaporator [22].

Furthermore, the highest total phenolic content obtained from leaves extract was 701 mg GAE/g extract using ethanol 95% maceration methods. It also showed great antioxidant activity with IC50 1.37 by ABTS assay [18]. Hardinsyah et al., 2019 reported antioxidant activity of *B. macrophylla* Griffith leaves in different maturity stages with several solvents (ethanol, ethyl acetate, and hexane). The result showed that the highest phenolic compound was 30.84 mg GAE/g, obtained from mature leaves with hexane extraction. Meanwhile, the ethanol extract of mature leaves generated the best antioxidant activity with a reducing power of 5.62 mg FeSO_4_ equivalent/g [21]. These reports concluded that the mature plum mango leaves showed good antioxidant activity, while the extraction technique to gain the most phenolic compound was maceration with 95% ethanol.

Besides seeds and leaves, the fruit of *B. macrophylla* Griffith has also been reported to have antioxidant activity. It was studied with different extraction solvents in the ripe and unripe fruit of *B. macrophylla* Griffith. The antioxidant activity of ripe and unripe *B. macrophylla* Griffith varied, but the unripe fruit of this plant showed better radical scavenging activity than the ripe fruit. As reported by Rajan and Bhat (2016) and Sukalingam (2018), unripe fruit with methanol extraction generated great DPPH scavenging activity of 70–77%. Meanwhile, the highest phenolic content was found in methanolic extract of unripe fruit with 50 mg GAE/g extract [2,19].

Phenolic compounds act as an antioxidant by reacting with various free radicals. The mechanism of antioxidant actions involve either hydrogen atom transfer, transfer of a single electron, sequential proton loss electron transfer, or chelation of transition metals. Moreover, as reported by Adam et al., 2021 antioxidant activity of plum mango extract is comparable to ascorbic acid as a potent antioxidant. Ascorbic acid acts directly to scavenge oxygen or nitrogen-based radical species generated during normal cellular metabolism [40]. The antioxidant mechanisms of ascorbic acid are based on hydrogen atom donation to lipid radicals, quenching of singlet oxygen, and removal of molecular oxygen [49]. The overall mechanisms of *B. macrophylla* Griffith as an antioxidant are summarized in Figure 2.

### 5.2. Anticancer

Cancer is a disease that is still a relatively high cause of death, and cases continue to increase every year. According to 2010 WHO data, cancer ranks second with the highest number of deaths after cardiovascular disease [50]. Cancer treatment with conventional therapy, such as chemotherapy surgery and radiation therapy, is not adequate for metastatic cancer [51]. The failure of cancer therapy is due to the resistance to treatment, including both chemotherapy and radiotherapy. *B. macrophylla* Griffith is believed to have the anticancer activity described in Table 2. The cytotoxicity assay used was methylthiazol2yl25-diphenyltetrazolium bromide (MTT) against Vero cells from the kidney of the African green monkey (*Cercopithecus aethiops*). These cells are homologous to human cells and are easy to grow [52]. The cytotoxicity test results of the ethanol extract of plum mango leaves on Vero cells showed a change in the morphology of Vero cells after the MTT test was carried out. This is because the proteins that play a role in cell adhesion do not polymerize. They interfere with cell adhesion, lipid membrane binding disrupts cells and induces cell apoptosis [23].

The ethanolic extract of plum mango leaves induced the mechanism of cell morphology changes induced by apoptosis. Morphological changes of Vero cells undergoing apoptosis include shrinkage of membrane vesicles, chromatin condensation, apoptosis determination, and nuclear fragmentation [53]. The cytotoxic effect of cytotoxins can cause changes in cell membrane permeability or impair the integrity of the cell membrane, rendering it non-viable and causing cell death [54]. Cell death is related to the cytotoxicity of a substance; this may be due to the biochemical mechanism of adenosine triphosphate (ATP) dilution and membrane defects [55]. 

Dehydrogenase is an enzyme that plays a role in the formation of ATP [56]. Inactivated dehydrogenase enzymes can produce a cytotoxic effect, resulting in a decrease in ATP where the cell’s functional activity is affected, and the cell dies [57].

Breast cancer is one of the most common cancers in women and significantly increases their morbidity and mortality rates [58]. Many advanced breast cancer treatment strategies are currently being developed in each country to increase the effectiveness of treatment and reduce cancer morbidity and mortality [59]. Unfortunately, the treatment goals for breast cancer have not been met in all patients because the remaining cancer cells, known as minimal residual disease (MRD), are defined as cells that can survive treatment leading to tumor recurrence and treatment failure. There is growing evidence that treatment failure associated with the use of cancer stem cells (CSC) represents a human population that is more resistant to radiation and resistant to other types of cancer [60].

Recent studies have demonstrated that tumor radiation impedance mechanisms are involved in several signaling pathways, including the epithelial–mesenchymal junction (EMT) binding to the adenosine triphosphate binding cassette (ABC) transporter and phosphatidylinositol3 kinase protein kinase B (PI3K/AKT) [61]. In determining the involvement of EMT in the differentiation of non-CSC into CSC and the occurrence of radioresistance, CSC phenotypes such as CD44^+^ and CD24^−/low^ were observed. In addition, the high expression of transporter-resistant proteins such as P-glycoprotein (P-gp), multidrug resistance-associated protein 1 (MRP1), and breast cancer resistance protein (BCRP) are involved in the resistance of breast cancer cells to drugs and radiotherapy. These proteins are encoded by the ABCB1, ABCC1, and ABCG2 genes, respectively [62].

Irradiation led to downregulation of ABCC1 and ABCG2 genes, but upregulation of ABCB1. MCF7 breast cancer cells that are thought to survive treatment with ionizing radiation (MCF-7/IR6) showed a chemoresistance phenotype by expressing high levels of MDR1 (ABCB1) but not of MRP1 (ABCC1) or BCRP (ABCG2). Overexpression of MDR1 with high Pgp pump activity and poor response to doxorubicin was found in MCF-7/IR6 compared to MCF-7 stem cells. Before irradiation, pretreatment of MCF-7 with plum mango seed extract (MPSE) affected the chemoresistance (MCF-7/MPIR6). MCF-7/MPIR6 cells decreased Pgp (ABCB1) mRNA expression but still expressed MRP1 and BCRP mRNA similar to parent MCF7 [27]. This shows that pretreatment with MPSE is able to prevent drug resistance (chemoresistance) by reducing the expression of the ABCB1 gene associated with the drug efflux pump (P-gp).

Gallotanin-rich extract of plum mango seeds is also reported as a radiosensitizer in overcoming radioresistance. PI3K/AKT and MAPK signaling pathways are survival pathways that, when activated, can protect cancer cells from the toxic effects of radiation, causing radioresistance [63]. MPSE was reported to be able to increase DNA damage caused by radiation, which was characterized by increased expression of γH2AX in response to DNA damage. This shows that the combination of MPSE therapy with IR can increase the radiosensitivity of breast cancer cells and increase radiation-induced DNA damage. This combination therapy also increases apoptosis higher than IR therapy alone [27].

MCF-7/FIR is breast cancer cells that survive after radiation, have potential as malignant cells characterized by a high migration rate compared to MCF-7 parent cells [24]. Irradiation can induce EMT characterized by upregulation of vimentin and ZEB1, which are mesenchymal properties of cells [64]. MPSE is able to inhibit irradiation-induced EMT-associated migration and invasion through downregulation of vimentin and ZEB1 and increase E-cadherin, which is an epithelial cell trait. Irradiation can also increase CD44*+*/CD24^−/low,^ which is a marker of breast cancer stem cells (BCSC). Pretreatment with MPSE showed a decreased expression of CD44*+*/CD24^−/low^. Furthermore, in the formation of mammospheres, namely aggregates of epithelial stem cells from breast tumors that can develop into CSCs, MPSE is able to inhibit the formation of mammospheres [24]. This suggests that pretreating MCF7 cells with MPSE before radiation shows increased sensitivity of MCF-7 cells through inhibition of IR-induced EMT and CSC formation.

The mechanism of action of MPSE in increasing the radiosensitivity of cancer cells is through the regulation of PI3K/AKT and MAPK signaling pathways. Both of these pathways are activated by IR and regulate cellular processes involved in radioresistance, including apoptosis, proliferation, and metastasis [65]. IR upregulated the expression of phosphorylated AKT (p-AKT), p-ERK1/2, and p-JNK in MCF-7 cells, whereas inhibition of the AKT signaling pathway sensitized MCF-7 cells to IR. The combination of MPSE therapy with IR was able to downregulate p-AKT, p-ERK1/2, and p-JNK [27]. Therefore, it can be concluded that MPSE increases the radiosensitivity of cancer cells through the regulation of PI3K/AKT and MAPK signaling pathways.

MPSE as a radiosensitizer was also tested on the head and neck squamous cell carcinoma (HNSCC) cell line. Gallotanin and its main bioactive compound, pentagalloyl glucose (PGG), may enhance the efficacy of radiotherapy in HNSCC by inhibiting IR-induced prosurvival signaling and enhancing the effects of IR-induced DNA damage. In addition, inhibition of IR-induced accumulation in the cancer stem cell population, which is responsible for radiation resistance in cancer, was followed by inhibiting the anti-apoptotic pathway and increased chemotaxis IR-induced cell death in HNSCC [26]. Molecular mechanism studies have shown that MPSE or PGG can enhance HNSCC radiation sensitivity by targeting cancer stem cells via attenuated STAT3 activation to overcome resistance and improve clinical outcomes for patients. STAT3 activation, which is known to enhance survival and proliferation signaling, has been shown to play an essential role in regulating stem cell angiogenesis in many cells, especially HNSCC [66]. To maintain CSC properties, STAT3 regulates the expression of some subsequent target proteins associated with lineage characteristics. Many proteins and CSC markers have been identified downstream of STAT3, such as CD44, CD133, ALDH1, and the CSC regulatory transcription factors (Oct and Sox2). In addition, the self-renewal properties of CSCs are dependent on STAT3 activation [67]. Highly active STAT3 is positively correlated with high-quality HNSCC tissue and promotes SCC self-renewal and HNSCC radiation resistance. Blocking STAT3 activation by specific inhibitors effectively suppresses globular tumor formation and reduces the number of ALDH CD44 cells, and further induces apoptosis in HNSCC [68]. Inhibition of STAT3 by MPSE or PGG was able to suppress CSCs in FaDu and CAL27 cells. For HNSCC cells, it has been shown that treating CSC-rich populations with MPSE or PGG or adhesive ball derivatives helps regulate phosphorylated STAT3 expression [26].

Besides PGG, other bioactive compounds contained in MPSE are ethyl gallate (EG) and gallic acid (GA). PGG and ethyl gallate-rich extract of plum mango seed were reported to induce apoptosis through the mitochondria-mediated pathway [25]. This is a solution to increase the efficacy of cancer therapy, which causes resistance to apoptosis due to chemotherapy drugs and radiation. Apoptosis via the mitochondrial-independent pathway (intrinsic pathway) is mainly triggered by non-receptor stimulation, including oxidative stress and DNA damage [69]. Mitochondria are the main source of reactive oxygen species (ROS), where excessive levels of ROS can cause mitochondrial dysfunction and induce cell death [70].

PGG induced S-phase and G0/G1 cell cycle arrest and apoptosis in breast cancer cells by inhibiting cyclin D1 and influencing specific apoptosis-related proteins such as Bax and Bcl-2 [71]. PGG can also inhibit triple-negative mammary gland growth and metastasis by inhibiting the JAK1-STAT3 signaling pathway and practicing antibiotic, anti-proliferative, and apoptotic induction [27]. EG has been shown to suppress the proliferation and invasion of breast cancer cells. Inhibition is regulated by the PI3K/AKT pathway, whereas EG treatment reduced the activity of substrate metalloproteinase 2 (MMP2) and MMP9 in MDAMB231 cells [72]. Apoptotic induction is regulated by changing the Bax/Bcl2 ratio [73]. GA has been shown to induce anti-proliferative and apoptotic activity in MCF-7 cells by increasing p27 levels, decreasing proliferation, and inducing cell cycle arrest in the G2/M phase [74].

Apoptosis via the mitochondria pathway is triggered by DNA damage or increased ROS. Free radical molecules can cause oxidative stress where there is an imbalance between tissue antioxidant capacity and ROS biosynthesis [75]. Excess ROS can irreversibly induce cell damage and death via intrinsic apoptotic pathways in mitochondria leading to dysfunctional injury and increased cell apoptosis in mitochondrial DNA [76]. Deoxyribonucleic acid (DNA) is damaged, causing p53 protein to accumulate in cells [77]. During this G1 phase, DNA repair occurs during the replication process, but if the damage involves stimulation of sensors that regulate apoptotic proteins, Bax and Bak, it causes proapoptosis [78]. Synthesis of the cl2 fraction of p53 protein triggers apoptosis [79].

Studies on the anticancer activity of *B. macrophylla* Griffith are dominated by the seeds. Dechsupa et al., 2019 reported that the main bioactive compounds of plum mango seed extract are gallotanin, EG, GA, and PGG, where these bioactive compounds act as anticancer agents [3]. These bioactive compounds work in several sites as anticancer, including DNA damage-induced ionizing radiation, regulation of cancer marker proteins such as ERK, AKT, STAT3, and p53 which ultimately induces the process of apoptosis. Moreover, phenolics and their derivatives work by denaturing cellular proteins in cell membranes. Denaturation of proteins in the cell membrane causes changes in cell permeability [80]. As a result, the cell membrane no longer holds the cell’s contents and blocks the flow of material into the cell, causing cell death [81].

According to the description of the molecular mechanism of anticancer activity of *B. macrophylla* Griffith above, the study was quite extensive, including how plum mango extract could be a candidate for new cancer therapies to overcome cancer therapeutic resistance such as chemoresistance or radioresistance by targeting CSCs as therapeutic targets. The mechanism of action of plum mango molecularly through various signaling pathways and regulation of protein markers lead to induction of apoptosis, as described in Figure 4. Gallotanin works by increasing DNA damage due to ionizing radiation, inhibiting AKT and ERK, which in turn induces apoptosis through the mitochondria-mediated pathway. PGG works by inhibiting STAT3 and EGF, thereby inducing apoptosis and stimulating p53 expression, which causes increased expression of p21 leads to cell cycle arrest.

### 5.3. Antimicrobial

Pathogenic microorganisms are the cause of various infectious diseases, especially infections in various tracts or body parts [82]. The biggest problem of its treatment is the resistance that occurs in pathogenic microorganisms to antibiotics. Cause less effective treatment and even lead to failure of therapy [83]. *B. macrophylla* Griffith or known as plum mango has been used as antibiotic by the community. Seeds and skins are usually underutilized, so they are referred to as waste products [84]. However, it has been widely reported that the seeds and skin of the fruit have high antioxidant value and contain many bioactive compounds. It could be a natural source of antioxidants that is easy to obtain and inexpensive [85]. Various fruit seeds (mango, longan, lychee, tamarind, rambutan, etc.) have been reported to contain potential medicinal properties that can be used as an antioxidant, anti-inflammatory, or antimicrobial [10,86,87,88]. The antibacterial properties of plum mango seed extract have been tested in vitro. Plum mango antimicrobial testing includes fungi, Gram-negative bacteria and Gram-positive pathogenic bacteria. Those species are *Candida albicans* (fungi), Gram-negative bacteria (*Eschericia coli*, *Vibrio parahaemolyticus*, *Shigella boydii*, *Shigella flexneri*, *Vibrio cholera*, *Pseudomonas aeruginosa, Salmonella enteritidis*, *Proteus mirabilis, Klebsiella pneumonia*, *Enterobacter aerogenes*) and Gram-positive bacteria (*Enterococcus faecalis, Staphylococcus aureus, Listeria monocytogenes, Streptococcus gordonii, Bacillus cereus)*.

*Candida albicans* is a fungus whose natural habitat is in the human body. Candida fungi are found in the digestive tract, mouth, vagina, rectum (anal canal), and other parts of the body that have warm temperatures [89]. When the number of *Candida albicans* in the body exceeds a reasonable limit, this risks causing dangerous infections that can spread to various organs of the body, such as the bloodstream, heart, kidneys, or brain [90]. Seed and leave extract plum mango has antimicrobial activity to this fungi, with the minimum inhibitory concentration (MIC) 0.21–0.25 mg/mL and minimum fungicidal concentration (MFC) 2.50 mg/mL [3,28]. Moreover, zone inhibition of plum mango seed against *Candida albicans* is 13.60 mm comparable to chlorhexidine with 15.8 mm (Leelapornpisid and Poomanee, 2021). It could be concluded that plum mango extract has potential antimicrobial against *Candida albicans*.

Pneumonia is an inflammatory process in the lung parenchyma and is a major cause of morbidity and mortality in children under five years of age, especially in developing countries. The causes are bacteria, viruses, fungi, exposure to chemicals or physical damage to the lungs, as well as indirect effects from other diseases [91]. Bacteria that can cause pneumonia include *E. coli, P. aeruginosa, S. aureus,* and *B. cereus* [92]. Seed plum mango extract exhibits antimicrobial activity against *E. coli, P. aeruginosa,* and *S. aureus* with MIC 312.5 µg/mL, 312.5 µg/mL, and 156.2 µg/mL, respectively. Seed plum mango extract also shows bactericidal to *S. aureus* with MBC 312.5 µg/mL [3]. Moreover, plum mango leave extract has an antimicrobial effect against *B. cereus* with an inhibition zone of 24.83 mm at 500 mg/mL and is comparable to ciprofloxacin as a control [28].

*Shigella* is a Gram-negative pathogen and the cause of shigellosis, a potentially deadly diarrheal disease whose symptoms range from mild intestinal discomfort to death depending on severity [93]. *Shigelle flexeneri* and *Shigella boydii* are of particular epidemiological importance in developing nations such as African and Asian countries [94]. Plum mango seed extract has an antimicrobial activity to *Shigella* with MIC 78.1 µg/mL but does not seem bactericidal effect [3]. Meanwhile, the inhibition zone of the leave plum mango extract is 22.5 mm at 500 mg/mL and not comparable to ciprofloxacin that was used as a control [28].

Salmonellosis is a disease caused by infection with *Salmonella* bacteria in the intestinal tract. This disease can be transmitted through food and drink contaminated with *Salmonella* bacteria, one of which is *S. enteritidis* [95]. The seed extract of plum mango elicit an inhibitory effect against *S. enteritidis* with MIC 520.8 µg/mL but have no bactericidal effect [3].

Listeriosis is a foodborne disease caused by the bacterium *Listeria monocytogenes*. Generally, the symptoms that appear are not severe, similar to those of influenza, such as fever, chills, back pain, headache, sometimes accompanied by nausea, vomiting, and diarrhea [96]. Plum mango leave extract shows antibacterial activity against *L. monocytogenes* with inhibition zone 17.83 mm at 500 mg/mL. It also shows the inhibition zone at the lowest concentration of 100 µg/mL with an inhibition zone of 11.5, however, this activity is incomparable to that of ciprofloxacin as a control with the inhibition zone against *L. monocytogenes* is 30.66 mm at 500 µg/mL [28].

Urinary tract infection (UTI) is a condition in which infection occurs in the organs included in the urinary system, namely the ureters, kidneys, bladder, and urethra. Often, the cause of urinary tract infections is the bacterium *Escherichia coli* (*E. coli*), which is found in the intestines [97]. However, this disease can also be caused by other types of bacteria such as *P. mirabilis*, *K. pneumonia*, and *E. aurogenes*. When *E. coli* bacteria are on the skin or near the anus, these bacteria can enter the urinary tract and move to other places [98]. Plum mango extract reported has a bacteriostatic and bactericidal effect against these bacteria that caused UTI. The seed extract exhibits antibacterial activity against *K. pneumonia* and *E. aurogenes* with MIC 520.8 µg/mL and 312.5 µg/mL, respectively, whereas, it shows bactericidal effect against *P. mirabilis* with MBC 1250 µg/mL and MIC 520.8 µg/mL [3].

Root canal is the term used to describe the natural cavity in the center of a tooth, whereas pulp is the soft part inside a tooth that contains blood vessels, nerves, and connective tissue [99]. Mixed bacterial–fungal biofilms are always present in the oral environment, including infected root canals. *E. faecalis*, *S. gordonii*, and *C. albicans* are the three most commonly recovered species in root canals undergoing retreatment due to failure of the primary endodontic treatment and with persistent infection [100]. The ethyl acetate seed extract of plum mango exhibits bactericidal (MBC) and candidical (MFC) at 2.50 mg/mL to *E. faecalis*, *S. gordonii*, and *C. albicans*. It also shows a significant inhibition zone against *E. faecalis* and *C. albicans* comparable to chlorhexidine used as a control [29]. It could be concluded that plum mango seed extract can be potentially used to treat root canal infections.

Furthermore, plum mango seed extract was also tested against several antibiotic-resistant bacteria, including methicillin-resistant *Staphylococcus aureus* (MRSA), extended-spectrum β-lactamase (ESBL)-producing *Escherichia coli* and vancomycin-resistant *Enterococcus faecalis*. MRSA is an infection caused by *Staphylococcus aureus* bacteria that can no longer be treated with various classes of antibiotics that are commonly used. *Staphylococcus* is actually bacteria that is not harmful and usually lives on the skin and nose of humans. However, if the growth is not controlled, these bacteria can cause various infections in the human body. *Staphylococcus* infections can generally be treated with antibiotics. However, as a result of decades of irrational use of antibiotics, a type of *Staphylococcus* has emerged, such as MRSA, which can no longer be treated with various commonly used antibiotics [101]. The ethanolic seed extract of plum mango show inhibition to drug-resistant bacteria with MIC 104.1 g/mL [3]. Furthermore, it also has an inhibitory effect on vancomycin-resistant *E. faecalis* (VRE), with MIC 78.1 g/mL [3]. VRE is a major health problem in many countries as VRE is a reservoir of glycopeptide resistance and is thought to be able to infect humans through contact (contact) with animals or eating (consumption) meat. Although *E. faecalis* infection is more common in humans, vancomycin resistance is more common in *E. faecium* isolates. VRE is a pathogen in immunocompromised populations, especially patients receiving various antibiotics and undergoing long hospitalizations. VRE is one of the causes of nosocomial infections, and its susceptibility (resistance ability) can be transferred between organisms or other species; thus, infection control policies (infection control) and guidelines for the administration of antibiotics are very important to be applied to control the spread of VRE and organisms that are susceptible (resistant) to various drugs [102].

Another important bacterial resistance is ESBL. ESBL is an enzyme produced in the plasmid of Gram-negative bacteria from the Enterobacteriaceae group that already has resistance to β-lactam antibiotics. The most commonly recognized ESBL-producing bacteria are *E. coli* and *K. pneumonia* and are often considered a major cause of UTI, pneumonia, and sepsis. These ESBL-producing bacteria are nosocomial pathogens and are increasingly being found as infectious agents in the community. There have been several studies showing an association between the transfer of *E. coli* or ESBL-producing ESBL genes from birds and pigs to humans who come into direct contact with these animals. In addition to direct zoonotic transfer, food of animal origin has the potential to be a risk factor for bacterial colonization or infection in humans [103]. Plum mango seed ethanolic extract has inhibitory activity against ESBL-producing *E. coli* with MIC: 520.8 g/mL [3].

The plum mango tree seeds (*B. macrophylla* Griffith) have antibacterial properties similar to those of the mango (*Mangifera indica* L.). Again, both belong to the Anacardiaceae family, and their properties are well documented. Plum mango seed extract has shown significant antibacterial activity against strains of Gram-positive bacteria, including *S. aureus*, *Bacillus* sp., *Clostridium* sp., and *Listeria monocytogenes*, with a MIC of about 50,500 g/mL [104]. The antibacterial activity of plant extracts, including plum mango seed extract, is known to be due to the presence of hydrolyzed tannins or galotans, which have nine galloil groups, flavonoids, and phenolic acids. Pentagloylglucopyranose is an important compound found in plum mango seeds [105,106,107]. In particular, the antibacterial activity of gelatonin increases directly with its molecular weight [108].

Terpenoids, alkaloids, and phenolic compounds reported inhibit the growth of perishable bacteria and foodborne bacteria, alter bacteria’s enzymatic activity, and damage bacteria microbial cell membrane proteins [109]. The plum mango leave components identified considered the most important for extract bioactivity are polyphenols, flavonoids, caryophyllene, phytols, and transgeranylgeraniol [110]. Based on these results, the ethanolic extract of the leaves can be a good candidate for studying natural antibacterial agents against infections or diseases caused by the tested microorganisms.

### 5.4. Antihyperglycemic

Hyperglycemia is a condition in which blood glucose levels are higher than usual. This long-term condition can lead to diabetes mellitus (DM) [111]. DM is a metabolic disease characterized by hyperglycemia due to insulin deficiency or insulin resistance [112]. The digestive system influences the body’s metabolism; in this process, several enzymes such as amylase and glucosidase are involved [113]. This enzyme plays a role in converting carbohydrates into simple sugar molecules. Therefore, interfering with these enzymes can increase the body’s metabolism [114]. In several studies reported, inhibition of both enzymes affects the control of hyperglycemia. *B. macrophylla* Griffith (plum mango) was reported to have inhibitory activity on a-amylase and a-glucosidase enzymes, summarized in Table 3.

Plum mango juice has glucosidase inhibition. In hydrolyzed juice extracts, this may be derived from myricetin and quercetin (previously reported as potent glucosidase inhibitors). High blood sugar (hyperglycemia) can be returned to normal (euglycemia) by activating carbohydrate-digesting enzymes such as glucosidase [15]. Glucosidases are known to catalyze the final step in carbohydrates into absorbable monosaccharides [115]. To date, only the volatile components of the plum have been identified. The main volatile components of this fruit are ocimene (E) (68.59%) and pinene (8.0%) [34]. This compound was also detected in the essential oil of black pepper (*Piper guineense*), which inhibited the glucosidase activity of *Phyllanthus acidus* extract, a weak glucosidase inhibitor found previously in two studies [116,117]. This may be due to the higher content of glucosidase inhibitors compared to the extract.

The metabolic enzyme that contributes to hyperglycemia is amylase. It is an enzyme that breaks down starch into simple sugars such as dextrin, maltotriose, maltose, and glucose [118]. Inhibition of amylase enzyme activity is an effective way to control blood sugar [119]. Therefore, the plum mango leaf extract with the highest amylase inhibition had the highest antioxidant activity, total phenol content (TPC), and total flavonoid content (TFC). The extract was prepared from plum mango leaves by the maceration extraction method [20]. Adequate blood sugar maintenance can be obtained from natural sources, such as bioactive compounds in plants that can act as amylase inhibitors [120]. Inhibitors of natural origin, such as vegetables that are rarely eaten, are very definite because they do not cause side effects when ingested [121]. In addition, Samudra et al. (2015) found that secondary metabolites, compared to primary metabolites, had a high ability to inhibit amylase enzyme activity [122]. Phenolic acids and flavonoids have been reported to have amylase enzyme activity [123]. Phenolic acids and flavonoids form covalent bonds with amylase to form quinones or lactones and react with nucleophilic groups, resulting in changes in amylase enzyme activity [124]. The reaction of phenolic compounds and proteins can inhibit the activity of enzymes, including amylase enzymes. 

Polyphenol complexes with starch so that the amylase enzyme cannot recognize the substrate [125]. Effects of flavonoids on amylase enzymes by enzyme kinetics and fluorescence spectroscopy have shown that flavonoids form complexes with amylase enzymes [126]. Flavonoids are hydrophobically linked to the enzyme amylase [127]. Flavonoids can also inhibit amylase activity because they can form quinones with oxopiran [128]. Flavonoids are antioxidants that can prevent the gradual decline in pancreatic cell function caused by oxidative stress [129]; this can reduce the incidence of type 2 diabetes and prevent long-term complications of diabetes [130].

Diabetes is often accompanied by postprandial hyperglycemia, a condition in which blood sugar levels rise excessively after eating [131]. This condition can cause an increase in glycated hemoglobin (HbA1C) and can also lead to complications of diabetes. It is essential to reduce postprandial hyperglycemia with oriental medicine to avoid such complications [111]. One possible mechanism for reducing postprandial hyperglycemia is by inhibiting the activity of glucosidase, an enzyme that hydrolyzes carbohydrates to glucose in the gut [128]. Inhibition of glucosidase activity can slow postprandial glucose uptake and ameliorate postprandial hyperglycemia [132]. Slow absorption of glucose from refined is one of the mechanisms used to reduce hyperglycemia in diabetes [133]. This can be achieved by inhibiting the glucosidase reaction in the small intestine. Glucosidase is an enzyme that is important for digesting carbohydrates. This enzyme is present in the membrane on the brush border surface of the upper small intestine [134]. This hydrolyzes the terminal glucose at the non-reducing end of the substrate, namely H. maltose or sucrose [135]. By activating glucosidase activity, less glucose is produced and absorbed in the bloodstream, reducing hyperglycemia, especially after eating [136]. 

The aqueous extract of plum mango was a potent glucosidase inhibitor with an IC50 value of 57.62 g/mL [40]. This suggests that plum mango extracts can slow down glucose absorption from the small intestine into the bloodstream, thereby reducing postprandial hyperglycemia. Overall possible mechanisms of antihyperglycemic effects of *B. macrophylla* Griffith described in Figure 5. However, the inhibitory effect of the extract was weaker than that of acarbose as a standard drug. Acarbose strongly inhibits glucosidase activity, as indicated by the low IC50 value (33 g/mL). This is possible because plum mango extracts are still raw and contain a mixture of bioactive and non-bioactive compounds. It is possible that the lower the concentration of biologically active compounds, the lower glucosidase inhibition. Unlike plum mango extract, acarbose is a nitrogenous pseudotetrasaccharide unit, and its glucosidase inhibitory effect has been well enhanced [137]. Similar to the antioxidant activity, the glucosidase inhibitory activity of plum mango seed extracts may be due to many phenolic compounds in the plant. This suggestion is consistent with previous studies showing that phenolic compounds contribute significantly to *Neptunia oleracea* glucosidase inhibition [138].

### 5.5. Antiphotoaging

The skin ages, causing roughness, dryness, sagging, and wrinkling. Ultraviolet B (UVB) rays cause skin diseases such as uneven pigmentation, sunburn, skin tanning, and skin cancer [139]. UVB stimulates the production of ROS, including hydroxyl radicals, peroxides, superoxides, and singlet oxygen [140]. Significantly increased levels of ROS cause damage to cellular structures and accumulation of oxidative stress in cells, which results in imaging phenomena [141]. *B. macrophylla* extract (BME) has possessed antiphotoaging and moisturizing effects on hairless mice in vivo study. Three parameters, including wrinkle, skinfold thickness, and elasticity were observed in dorsal mice. BME treatment shows the ability to prevent wrinkle, skinfold thickness, and abnormal elastic fiber in dorsal mice compared with induced-UVB without BME treatment.

Increased oxidative stress induced by UVB light plays a central role in imaging through activation of MAPK and nuclear factor kappa B (NFκB), proteins necessary for inflammation [142]. Increasing the expression or activation of the antioxidant enzyme catalase is a vital strategy to protect the skin from photooxidative stress [143]. Activated NFκB stimulates the release of inflammatory cytokines such as interleukin 6 and interleukin 8, which induce matrix metalloproteinase (MMP) expression [144]. MMPs are a response to the degradation of collagen, gelatin, and extracellular matrix components (ECM). BME is able to inhibit the MAPK signaling pathway, which further inhibits NFκB, thereby suppressing the expression of MMPs. With a decrease in MMPs, collagen degradation decreases and maintains collagen levels in the skin.

Collagen is an essential component of the dermis and plays an essential role in wrinkling [145]. Type I and type III collagen are the most common types of collagen found in the extracellular matrix (ECM) and connective tissue [146]. Type IV collagen is found only in the basement membrane and, together with type VII collagen, plays a vital role in the adhesion of the skin epidermis [147]. Gelatinase (MMP2 and MMP9) digests type IV collagen (which plays an essential role in epidermal-dermal adhesion) and gelatin [148]. Upregulation of collagen synthesis gene expression leads to an increase in hydroxyproline content, a marker of collagen fibers [149]. It can be concluded that the hydroxyproline content increased significantly in the plum mango extract treatment. Transforming growth factor (TGFβ) binds to its receptor, causing Smad2/3 activation and stimulating type I procollagen synthesis [150]. Smad7 acts as an antagonist of the TGFβ/Smad signaling pathway by inhibiting Smad2/3 activation [151]. MMP accelerates wrinkle formation in response to the breakdown of collagen, gelatin, and other ECM components [152]. In addition, MMP3 introduces a large number of ECM components and promotes the activation of other proMMPs [153]. MMP expression and secretion are regulated by the MAPK/AP1 complex signaling pathway [154].

Excessive oxidative stress induces activation of MAPK signaling components, resulting in JNK phosphorylation [155]. Activated JNK stimulates cFos protein expression and cJun phosphorylation, leading to the formation of the AP1 complex [156]. Corneal cells are late differentiated keratinocytes that are responsible for the formation of the stratum corneum (SC) and the production of natural hydration factor (NMF) [157]. CE is formed by cross-linking different proteins such as loricrin and involucrin. Loricrin is the main component of CE, and involucrin plays a significant role in CE formation. Transglutaminase participates in CE formation by stimulating cross-linking between loricrin and involucrin. When filaggrin is degraded by caspase1, NMF is formed, which is a determinant of hydration. Profilaggrin is broken down into filaggrin by profilaggrin-degrading enzymes such as matriptase and prostasin [158]. The expression levels of matriptase and prostacyn proteins were increased by treatment with plum mango extract in hairless mice induced by UVB light. Previous studies have shown that activation or inhibition of Smad2/3 of ERK and JNK, MAPK members, stimulates keratinocyte differentiation, which is required for EC formation and skin hydration. Activated TGFβ/Smad signaling pathway, inactivated MAPK/AP1 signaling pathway, or a combination of the two signaling pathways are also involved in the UVB-induced moisturizing effect of plum mango extract on mouse skin.

Overall, BME shows its antiphotoaging effect, which prevents wrinkle formation in UVB-irradiated hairless mice by downregulating the signaling pathway of the MAPKs/AP1 combination complex, which regulates MMP expression shown in Figure 6. BME increased collagen mRNA expression and stimulated the TGFβ/Smad signaling pathway in response to the regulation of the AP1 complex, resulting in increased collagen content in the dermis. In addition, plum mango extract helps moisturize the skin by stimulating EC formation and filaggrin processing. Thus, *B. macrophylla* Griffith could be a potential anti-aging and moisturizing agent in skin supplements. Further clinical research is needed to determine whether BME is clinically effective as a natural moisturizer and anti-inflammatory supplement.

## 6. Improve Vegetable Intake and Blood β-Carotene Concentration

Many components in fresh vegetables can affect skin quality, such as nutrients and phytochemicals, including amino acids (NA-acetylcysteine), carotenoids (β-carotene, lutein, zeaxanthin, and lycopene), atty acids (Linoleic acid, Eicosapentaenoic acid, and Docosahexaenoic acid), vitamins (vitamin C, vitamin E, and vitamin B3), minerals (copper, selenium, and zinc), and polyphenols [159,160]. Some of its ingredients are useful as antioxidants and skin pigmentation cofactors to protect the skin from damage, improve skin security and heal wounds. The most observable indicator of skin quality is skin luminosity. Among the components present in vegetables, carotenoids have the most significant influence on skin color. Carotenoids are one of the most abundantly found in plants, blood, and human tissues [161]. The primary function of carotene is to activate provitamine A. Carotene also acts as an antioxidant in the human body and on skin quality [162]. The concentration of carotenoids in the blood (including carotenes) is related to the concentration of carotenoids in the skin. Consumption of carotenoids increases the concentration of carotenoids in the blood and skin [163].

## 7. Conclusions

Plum mangos have been used as folk medicine. Due to its antioxidant, antibacterial, and anticancer activity, it may prove effective to test plum mango as a natural preservative in the food industry. Almost all parts of the plum mango have good pharmacological activity; the community often uses the leaves and fruit to make salads and juices, which are very beneficial for improving body health. Interestingly, from all parts of the *B. macrophylla* Griffith, the seeds, which were rarely used, actually showed good pharmacological activity. Its antioxidant activity shows an IC50 value comparable to a potent antioxidant, namely ascorbic acid. The anticancer activity of *B. macrophylla* Griffith seed extract has also been explored to the molecular level and its association with the main bioactive component, gallotanin. This discovery is excellent because the use of waste-product of the plant can be a source of new herbal drug candidates so that it is related to their economic value. However, exploratory data on this plant are still limited, starting with the lack of data on the bioactive compounds from each part of the plant and very little data on in vivo and clinical trials. Therefore, further research is needed to support the potential of this plant as an herbal plant. Regarding its pharmacological activity and potential for candidate novel drugs to improve the treatment of various diseases, the scientific community needs to develop further the benefits of this plant, such as toxicological studies and clinical trials.

## Figures and Tables

**Figure 1 pharmaceuticals-15-00238-f001:**
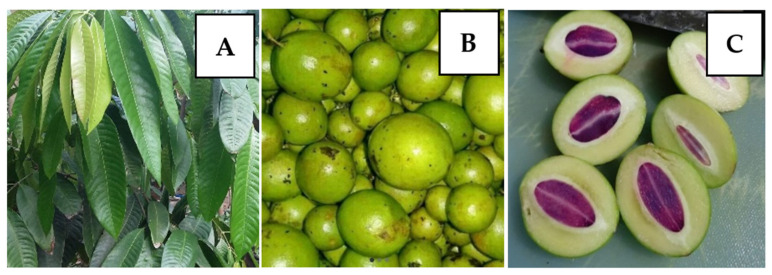
*Bouea macrophylla* Griffith; (**A**) leaves, (**B**) fruits, (**C**) seeds.

**Figure 2 pharmaceuticals-15-00238-f002:**
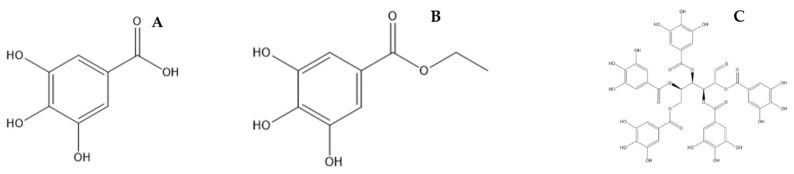
Chemical structure of gallic acid (**A**), ethyl gallate (**B**), and pentagalloyl glucose (**C**).

**Figure 3 pharmaceuticals-15-00238-f003:**
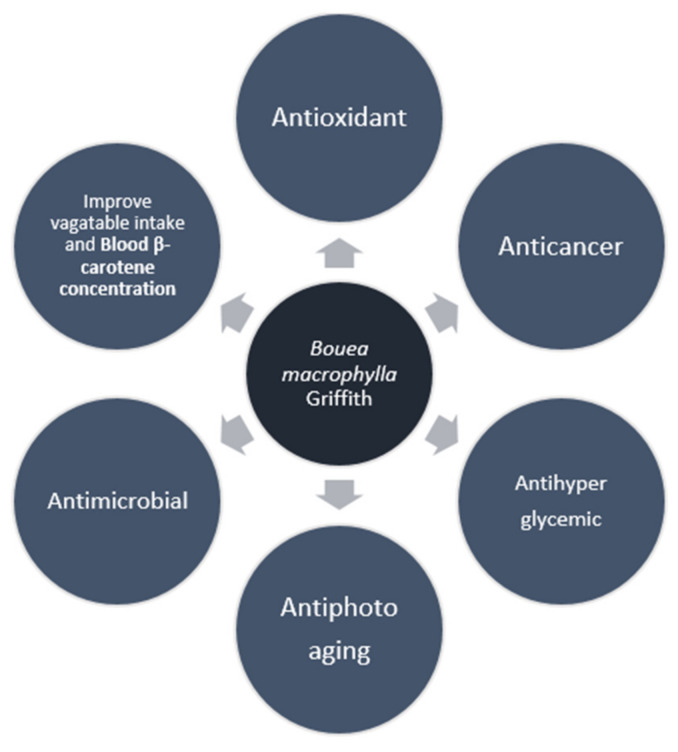
A summary of the pharmacological activities of *B. macrophylla* Griffith.

**Figure 4 pharmaceuticals-15-00238-f004:**
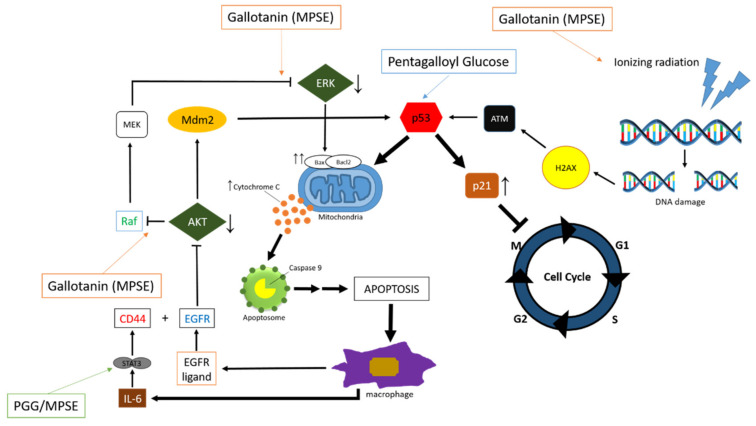
Anticancer molecular mechanisms of *B. macrophylla* Griffith.

**Figure 5 pharmaceuticals-15-00238-f005:**
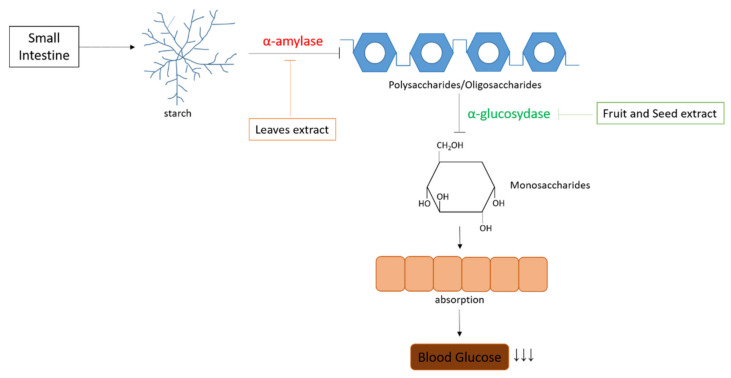
Antihyperglycemic mechanisms of *B. macrophylla* Griffith.

**Figure 6 pharmaceuticals-15-00238-f006:**
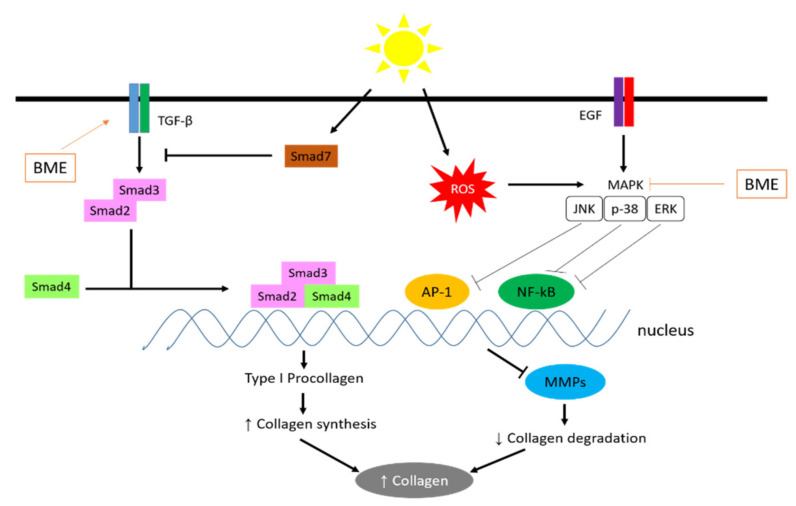
Mechanisms of action of *B. macrophylla* Griffith as an antiphotoaging.

**Table 1 pharmaceuticals-15-00238-t001:** Antioxidant Activity of *B. macrophylla* Griffith.

Plant Part		Extraction Technique	Total Phenolic Content	Antioxidant Assay	Antioxidant Activity	Reference
Seed (unripe and ripe)		Ethanol maceration	0.77 mg GAE/mg (unripe)	FRAP	114.98 μg Fe^2+^E/μg	[3]
				TEAC	2.21 μg TE/μg	
				DPPH	20.87 µg/mL	
			0.47 mg GAE/mg (ripe)	FRAP	94.82 μg Fe^2+^E/μg	
				TEAC	1.72 μg TE/μg	
				DPPH	31.14 µg/mL	
Seed		Chloroform maceration	686.04 mg GAE/g	DPPH	4.34 µg/mL	[40]
Leaves		Ethanol maceration	530.85 mg GAE/g	ABTS	1.37 µg/mL	[18]
				FIC	1.70 µg/mL	
Leaves		Ethanol maceration	68.53 mg GAE/g	DPPH	55.83 µg/mL	[17]
Leaves		Vacuum Evaporator extraction	117.836 mg GAE/g	DPPH	26 µg/mL	[22]
Leaves		Ethanol maceration	20 mg GAE/g	FRAP	5.62 ± 0.38 mg FeSO_4_ equivalent/g	[21]
		Hexane maceration	30.84 mg GAE/g		4.5 mg FeSO_4_ equivalent/g	
Leaves		Water maceration	364.56 mg GAE/g	DPPH	35 µg/mL	[20]
Fruit	Ripe	Aqueous maceration	-	DPPH	83%	[19]
	Unripe				82%	
Leaves					76%	
Fruit		Water maceration	372.35 µg GAE/g	FRAP	133.31 μg TEAC/g	[15]
				DPPH	258.17 μg VCEAC/g	
Fruit		Maceration (methanol, Ethanol, and distilled water)	-	FRAP	16,290.91 µM Fe(II)/100 g	[2]
				DPPH	77.69%	
				ABTS	99.76%	

**Table 2 pharmaceuticals-15-00238-t002:** Anticancer activity of *B. macrophylla* Griffith.

Plants Part	Extraction Techniques	Anticancer Assay	Cell Lines	IC50	Reference
Leaves	Ethanol maceration	MTT assay	Vero Cell	35.808 µg/mL	[23]
Seed	Ethanol maceration	MTT assay	Doxorubicin-sensitive and resistant leukemic(K562, K562/ADR) and lung cancer (GLC4 and GLC4/ADR) cells	4–16 µg/mL	[3]
Seed	Ethanol maceration	MTT assay	MCF-7/IR6 cells	215.42 nM	[24]
Seed	Ethanol maceration	MTT assay	MCF cells	6.94 µg/mL	[25]
Seed	Ethanol maceration	Mammosphere formation assay	Breast cancer stem cells (CSCs)		[27]
Seed	Ethanol maceration	Tumorsphere formation assay, colony formation assay, and apoptosis assay	Head and neck squamous cell carcinoma (HNSCC)	14.52 µg/mL	[26]

**Table 3 pharmaceuticals-15-00238-t003:** Antihyperglycemic activity of *B. macrophylla* Griffith.

Plants Part	Extraction Techniques	Antihyperglycemic Assay	Dose/IC50	Mechanisms of Action	Reference
Seed	Decoction	α-Glucosidase Inhibitory Assay	0.55 mg/mL	Delayed glucose absorption in the small intestine	[40]
Leaves	Water maceration	α-Amylase Inhibitory Assay	60 µg/mL	Inhibit α-Amylase by forming a complex	[20]
Fruit	Water maceration	α-Glucosidase Inhibitory Assay	83.44%	Delayed glucose absorption in the small intestine	[15]

## Data Availability

All data is included in the article.

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
