# Peer review of "The Evolution of Pharmacological Activities Bouea macrophylla Griffith In Vivo and In Vitro Study: A Review"

_pharmaceuticals, 2022, doi:10.3390/ph15020238_

Round 1

Reviewer 1 Report

The review under the title "Evolution of Pharmacological Activities of Bouea macrophylla Griffith In Vivo and In Vitro Studies: a review", is well written with a suitable title that can be accepted after covering the following points:

  • The introduction just mentioned data about the plant under study, it should be of a wider scope. More information should be added about the Family of the plant, its importance, history, distribution, and some of its important members with summarized data about each.
  • At the end of the introduction, the authors should add their aim from offering this review, why they have chosen this plant in particular (ex: due to its wide use in folk medicine in a special area, famous for a certain activity, an edible plant with biological importance,........etc.).
  • Section 4: "Phytochemical Constituent of Bouea macrophylla Griffith", upon search it was found that it lacks many published data of active constituents isolated from the seeds should be added with their references.
  • Line 122: It was mentioned "Bouea macrophylla Griffith reported has high phenolic compound", Why these phenolic compounds were not mentioned in Section 4 the phytochemical constituents. Add in details with the names and structures of these compounds.
  • Line 124: "The antioxidant activity of Bouea macrophylla Griffith has been explored by any plants". What is meant by this phrase?
  • Line 125: mentioned "This species’ seed, leaves, and fruit have antioxidant activity", a reference should be added to confirm this information.
  • Figure 2 is unclear. What does it add to the presented data (better removed).
  • In the conclusion section: Line 703 the authors mentioned "Plum mangos appear to be an excellent source of novel compounds'' than in Line 716 mentioned " the lack of data on the bioactive compounds from each part of the plant ". How comes? the 2 phrases are opposite to each other. Does the plant offer several biologically active compounds or the compounds are still unexplored and need to be studied further?
  • At the end of the conclusion add recommendations from your point of view concerning the plant and its possible benefits.

Author Response

dear Reviewer 1

Thank you so much for your kind suggestion and your kind advice  to our review article regarding our article with entitle : Evolution of Pharmacological Activities of Bouea macrophylla Griffith In Vivo and In Vitro Studies: a review. We believe your advice lead us to create good  quality article

Author’s response on the reviewer and editor’s remarks

All changes in the manuscript are written in yellow highlight text.

1st Reviewer

Comments and Suggestions for Authors

The review under the title "Evolution of Pharmacological Activities of Bouea macrophylla Griffith In Vivo and In Vitro Studies: a review", is well written with a suitable title that can be accepted after covering the following points:

  • The introduction just mentioned data about the plant under study, it should be of a wider scope. More information should be added about the Family of the plant, its importance, history, distribution, and some of its important members with summarized data about each.
  • Author response:

More information about  plants has been added in the text and colored with yellow highlight text

  • At the end of the introduction, the authors should add their aim from offering this review, why they have chosen this plant in particular (ex: due to its wide use in folk medicine in a special area, famous for a certain activity, an edible plant with biological importance,........etc.).
  • Author response:

More information about  the reason why we explore this plants has been added in the text and colored with yellow highlight text

  • Section 4: "Phytochemical Constituent of Bouea macrophylla Griffith", upon search it was found that it lacks many published data of active constituents isolated from the seeds should be added with their references.
  • Author response:

The references regarding  active constituents isolated from the seeds has been added accordingly

  • Line 122: It was mentioned "Bouea macrophylla Griffith reported has high phenolic compound", Why these phenolic compounds were not mentioned in Section 4 the phytochemical constituents. Add in details with the names and structures of these compounds.
  • Author response:

The names and structures of these compounds has been added accordingly

  • Line 124: "The antioxidant activity of Bouea macrophylla Griffith has been explored by any plants". What is meant by this phrase?
  • Author response:

It should be “.. by any part of the plants” means it has been explored in seeds, leaves, and fruits.

  • Line 125: mentioned "This species’ seed, leaves, and fruit have antioxidant activity", a reference should be added to confirm this information.
  • Author response:

The reference has been added accordingly.

  • Figure 2 is unclear. What does it add to the presented data (better removed).
  • Author response:

The figures have been deleted accordingly.

  • In the conclusion section: Line 703 the authors mentioned "Plum mangos appear to be an excellent source of novel compounds'' than in Line 716 mentioned " the lack of data on the bioactive compounds from each part of the plant ". How comes? the 2 phrases are opposite to each other. Does the plant offer several biologically active compounds or the compounds are still unexplored and need to be studied further?
  • Author response:

According to the data of this plant, the compounds still unexplored and need to be studied further. We’ve corrected the first sentence of the conclusion.

  • At the end of the conclusion add recommendations from your point of view concerning the plant and its possible benefits.
  • Author response:

The conclusion has been corrected accordingly.

Reviewer 2 Report

The authors present a study on evolution of pharmacological activities of Bouea macrophylla Griffith. The manuscript is well structured: macroscopic descriprtion, chemical content and pharmacological activities of different botanical parts of the investigated plant are described in details.  

I recommend the manuscript to be published after a minor revision.

My suggestions are:

  • It will be better to remove Figure 6 from the Conclusion section and put it in section 5. Pharmacological Activities of Bouea macrophylla Griffith;
  • The references included in the tables (Table 1, Table 2, Table 3) should be presented only as numbers without the names of the authors;
  • The authors could reduce the general information for some of the pharmacological mechanisms described in subsections 5.1-5.5. For example, in the subsection 5.1., lines 105-119, the general information about oxidative stress and antioxidants is too much. The focus should be only on the specific activities of Bouea macrophylla.

Author Response

dear Reviewer 2

Thank you so much for your kind suggestion and your kind advice  to our review article regarding our article with entitle : Evolution of Pharmacological Activities of Bouea macrophylla Griffith In Vivo and In Vitro Studies: a review. We believe your advice lead us to create good  quality article

Author’s response on the reviewer and editor’s remarks

All changes in the manuscript are written in yellow highlight text.

2nd Reviewer

Comments and Suggestions for Authors

The authors present a study on evolution of pharmacological activities of Bouea macrophylla Griffith. The manuscript is well structured: macroscopic description, chemical content and pharmacological activities of different botanical parts of the investigated plant are described in details. 

I recommend the manuscript to be published after a minor revision.

My suggestions are:

  • It will be better to remove Figure 6 from the Conclusion section and put it in section 5. Pharmacological Activities of Bouea macrophylla Griffith;
  • Author response:

We are agreeing with the advice from referee, we move the figure 6 and put in section 5

  • The references included in the tables (Table 1, Table 2, Table 3) should be presented only as numbers without the names of the authors;
  • Author response:

More information about citation write methods  has been changed only number of references write methods

  • The authors could reduce the general information for some of the pharmacological mechanisms described in subsections 5.1-5.5. For example, in the subsection 5.1., lines 105-119, the general information about oxidative stress and antioxidants is too much. The focus should be only on the specific activities of Bouea macrophylla.
  • Author response:

We already reduce the general explanation of following the suggestion from referee